# Australian Women Veterans’ Experiences of Gendered Disempowerment and Abuse Within Military Service and Transition

**DOI:** 10.3390/ijerph22040584

**Published:** 2025-04-08

**Authors:** Sharon Lawn, Elaine Waddell, Louise Roberts, Pilar Rioseco, Tiffany Beks, Liz McNeill, David Everitt, Tiffany Sharp, Dylan Mordaunt, Amanda Tarrant, Miranda Van Hooff, Jon Lane, Ben Wadham

**Affiliations:** 1College of Medicine and Public Health, Flinders University, Adelaide, SA 5042, Australia; elaine.waddell@flinders.edu.au (E.W.); louise.roberts@flinders.edu.au (L.R.); liz.mcneill@flinders.edu.au (L.M.); dylan.mordaunt@wfa.org.nz (D.M.); ben.wadham@flinders.edu.au (B.W.); 2Open Door Initiative, Flinders University, Adelaide, SA 5000, Australia; tiffany@atlaspsychologyconsulting.com (T.B.); djeveritt@telstra.com (D.E.); amanda.tarrant2@sa.gov.au (A.T.); 3Lived Experience Australia, Adelaide, SA 5000, Australia; 4School of Public Health, Queensland University of Technology, Brisbane, QLD 4000, Australia; mariapilar.riosecolopez@qut.edu.au; 5Australian Institute of Family Studies, Melbourne, VIC 3000, Australia; 6Werklund School of Education, University of Calgary, Calgary, AB T2N 1N4, Canada; 7College of Nursing and Health Sciences, Flinders University, Adelaide, SA 5042, Australia; 8Defence Force Welfare Association SA Branch, Adelaide, SA 5000, Australia; 9Cambrian Executive, Adelaide, SA 5000, Australia; 10Wellington Free Ambulance, Wellington 6140, New Zealand; 11Southern Adelaide Local Health Network, Adelaide, SA 5000, Australia; 12Veterans SA, Adelaide, SA 5000, Australia; 13Gallipoli Medical Research, Brisbane, QLD 4000, Australia; miranda@drvanhooffconsultancy.com.au; 14Military and Emergency Services Health Australia (MESHA), Adelaide, SA 5000, Australia; 15School of Medicine, University of Tasmania, Hobart, TAS 7000, Australia; jonathan.lane@utas.edu.au; 16College of Education, Psychology and Social Work, Flinders University, Adelaide, SA 5042, Australia

**Keywords:** women, veterans, gender, military sexual trauma, abuse, transition, identity, culture, systems, mental health

## Abstract

Disempowering experiences of military service and transition for women veterans exist within an established, dominant, masculinised culture, in which their presence is highly visible, challenged, and often subject to institutional prejudice. Sexual abuse of women in the military, in particular, is a persistent finding in contemporary international research and national inquiries into military culture in countries such as Australia, the United Kingdom (UK), the United States (US), and Canada. This study sought to understand military service, transition to civilian life, and post-military experiences of Australian women veterans, specifically their experiences of discrimination, military sexual harassment and assault, and consequent military sexual trauma (MST). In-depth qualitative interviews were undertaken with 22 Australian women veterans that examined how women veterans manage their identity as women in the military. Issues included gender-based challenges in conforming to a masculinised culture, experiences of misogyny, sexual harassment and assault, systemic failures to recognize women’s specific health needs, and experiences of separation from the military and transition, including help-seeking and engagement with services to address their experiences of MST. Women veterans’ adverse experiences largely stemmed from an entrenched masculinised military culture, in which military sexual assault was enabled, ignored, and condoned. Military and veteran support services have been slow to recognize, acknowledge, and address this significant issue.

## 1. Introduction

Women make up approximately one in five or less of currently serving military personnel across the Five Eyes countries (Australia—20.7% [1], Canada—16% [2], New Zealand—18% [3], United Kingdom—11% [4] and United States—17.5% [5]), and between 10 and 28% of veterans who have previously served in the military in these countries [6,7,8,9,10]. According to the Australian Defence Force (ADF) 2024 Annual Report [1], 20.7% of Australia’s 100,000 military personnel identified as women (female), whilst women make up 13.5% of Australia’s veterans (those who have served in the military and have been discharged or left to return to civilian life) [6]. However, the actual number of women veterans is understood to be higher than reported because some choose not to identify with their military service upon separating from the military [11]. In Australia, a veteran is any person who has served in the military for at least one day. Within the Australian military, women are employed across a full spectrum of roles alongside men; they conduct operations and deployments in order to defend Australia’s national interests and promote regional and global security and stability through their roles in the Navy (24.1%), Army (15.3%), or Air Force (27%) and are employed as either permanent Defence Force, Reservists, or Civilian staff [1]. They are, however, more likely to be in certain roles in Australia’s military; they make up 48.5% of the 19,465 Civilian staff, 18.6% of its 32,000+ Reservists, and 20.4% of its 57,226 Defence Force personnel [1]. Appendix A provides a brief chronological summary of how women’s involvement in the Australian military has been shaped over time.

Military service occurs in a male-dominated environment and culture in which women’s presence is highly visible, challenged, and often subject to institutional prejudice [11,12,13,14]. Challenges for women veterans in integrating into, and remaining accepted in, a masculinised military culture are a common theme in contemporary research internationally [12,15,16,17,18,19]. Acker [20,21] described similar experiences of barriers and challenges for women serving within other fields, such as the police, which exemplifies a highly regimented and traditionally male-dominated profession that has also drawn heavily from a military institutional structure and culture.

Women veterans experience gendered harassment, discrimination, and military sexual abuse, with consequent military sexual trauma (MST) commonly during their service, and this can have significant adverse impacts on their transition from the military to civilian life [1,22]. For example, the Australian Human Rights Commission (AHRC)’s 2014 inquiry into the treatment of women in the ADF (the Broderick Review) [23] found that women constantly face low-level sexual innuendo and misconduct and must defer to, resist, or embody the military masculinity culture to sustain themselves; however, this embodiment of the culture may lead them to avoid seeking help or disclosing MST during and after military service [24]. A further review into the treatment of women in the ADF, conducted by the AHRC in 2013–2014 [23], assessed the experiences of women across the Army, Navy, and Air Force, providing detailed insights into the ADF’s culture, policies, and practices concerning gender. The report offered five key principles, with several recommended actions within each, aimed at fostering a more inclusive and equitable environment for all service members. These included strengthening leadership to drive reform; diversity of leadership to increase capability; increasing opportunities for women; increase flexible work arrangements; and recognition that “Gender based harassment and violence ruins lives, divides teams and damages operational effectiveness” [23] (pp. 13–16). Despite these reviews and efforts stemming from their recommendations, a 10-year review (2013–2022) of key performance indicators for women serving in Australia’s military [11] (p. 63) found that, “Experiences of unacceptable behaviour have remained consistent, and consistently higher, for women than men since 2013. Over the same period men’s experiences of unacceptable behaviour has declined”; that is, little change in rates was recorded across the 10 years for women in the Navy (from 56% to 53%) and Army (52% to 53%), with some improvement for women in the Air Force (57% to 46%).

The 2024 final report on outcomes of the Australian Royal Commission in Defence and Veteran Suicide (RCDVS) found that ex-serving women discharged involuntarily for medical reasons are 5.2 times more likely to die by suicide compared to those discharged voluntarily [22,25]. The Australian government’s primary data repository, the Australian Institute of Health and Welfare (AIHW) [26], also reported elevated suicide rates for ex-serving ADF females (between 1997 and 2021) as 107% higher than the general Australian female population. Further, the 2024 RCDVS report (Volume 3) on military sexual violence, unacceptable behaviours, and military justice stated, “In the Australian Defence Force (ADF), the majority of sexual violence is perpetrated by men, and the majority of victims are women. We acknowledge that men are also victims and women are also perpetrators, but these gendered patterns cannot be ignored—particularly in the context of ex-serving women dying by suicide at twice the rate of the general female population.” [25] (p. 3). Of equal concern, that report also found that, “At the time of writing, the ADF still could not accurately quantify the prevalence of all forms of sexual misconduct in the workplace. Most managers and commanders have not undertaken dedicated training to respond to reports of sexual misconduct. There is no specific return to work policy for victims of sexual misconduct, and personnel systems are not designed to ensure victim safety”.

### Existing Research on Women Veterans’ Experiences of Gendered Disempowerment and Abuse Within Military Service and Transition

Sexual abuse of women in the military, in particular, is a persistent finding in the contemporary international research [12,15,16,17,27,28]. An Australian study [14] found female service members often do not report MST for fear of further victimisation, blame, and trauma. A US longitudinal study with more than 500 Iraq and Afghanistan War women veterans found that deployment sexual harassment was the strongest predictor of decreased psychosocial functioning across all domains [29]. A Canadian grounded-theory study with 20 women veterans [12] described how they face regular and persistent low-level sexual innuendo and misconduct and must defer to, resist, or embody masculine military culture to sustain themselves during service and in shaping their identity in transition to civilian life post-military service. These normalised experiences within a dominant gendered military culture may lead women veterans to avoid seeking help or disclosing MST during service and post-release from the military [24].

Eichler [16] conducted in-depth interviews and focus groups with 32 Canadian women veterans, exploring how gender and sex shape their military-to-civilian transition, finding that they are rarely recognised as veterans after their military service, they do not fit military or civilian gender norms, and they encounter systems set up on the assumption of veterans being men. Eichler [16] (p. 39) found that, “They were assigned menial tasks or kept out of the loop because they were women or had recently returned from maternity leave…approximately half the women reported facing gender-based violence during service in the form of ongoing harassment, physical abuse, or sexual assaults”. Several participants described ongoing physical and mental health conditions arising from this treatment during service, with MST survivors expressing feelings of deep betrayal by their peers and the institution because of how these incidents were handled [16].

Bourke [28] (p. 87) explored rape in branches of the U.S. military from the 1990s to 2020s, in particular, “the denial and dismissal of sexual violence in the U.S. military and the pathologization of victims; systemic failures of the US military institution and its culture.” Daphna-Tekoah et al. [28], citing evidence from US studies, reported that estimates of sexual assaults during military service in the US range from 9.5 to 49% among women. Dodd [30] reported similarly high rates of gendered violence and harassment experienced by women serving in the British Army; approximately 75% had been subjected to inappropriate and unwelcome comments; 20% had experienced inappropriate sexual touching; 8% had experienced a serious sexual assault; and 3% reported being raped.

There is significant and growing international research attention on the military–civilian transition [31,32], which is understood as a challenging period that can last for many weeks, months, and sometimes years for military veterans to adjust back to civilian life post-military service. Confusion in navigating a new identity and role post-service can arise for many reasons. A significant proportion of individuals enter military service in their late teens and early adulthood, which is a crucial period for identity formation, including sexual identity. The military becomes their ‘family’, and the military culture shapes the dominant set of codes by which they see themselves and others [12,33]. For women veterans, however, they fit neither in the military, due to gendered relations centred on military masculinity, nor in civilian life, where they are largely invisible as ‘veterans’; this ‘no women’s land’ is poorly understood [34], see also [13,14,35]. Challenges for women veterans are also evident within existing veteran transition and support services in the community. Support services are perceived as inherently gender-biased and have been developed largely for males and therefore fail to understand or address the needs of women veterans [14,36]. Within such limited support structures, there is a lack of disclosure and reporting of MST. This results in a lack of engagement with health services to deal with the impacts of MST on physical and mental health. This community service experience may result in and mirror experiences during service where MST was not reported (or not believed or was ignored when it was reported) once women veterans are discharged or leave the military.

Little is known about women veterans’ transition to civilian life after military service: why some women veterans successfully navigate transition while others continue to struggle [37,38]. Little is known about why rates of veteran suicide are high for women veterans [26] or about the mental health needs of women veterans [39] and what role experiences of discrimination, abuse, and MST during military service might play in contributing to these issues. Even less is known about what role dominant-masculinity military cultures might play for women veterans’ mental health and wellbeing. However, we do know that few programmes for transitioning veterans are effective for women veterans [12,36]. Few studies have investigated the experiences of Australian women veterans, specifically their experiences of discrimination and MST.

## 2. Materials and Methods

This paper examines experiences of gender and abuse that were reported as part of a larger study [34] that used a qualitative research methodology to explore the military service and transition experiences of Australian women veterans. The aim of the larger study was to better understand the impact of military service culture on transition and post-military service needs of Australian women veterans, to inform policies and programmes designed to support them. The study included a dedicated Veteran Project Reference Group (VPRG) of seven women and men, drawn from the community, who identified as veterans who had all served across various eras in the ADF. The VPRG was an established group within the University’s veterans and families Research Centre (Open Door) in which this research took place. It informed the study from its inception, across all aspects of study design, funding application, recruitment, data collection, data analysis and report writing, publication, and dissemination of findings to veteran community stakeholders.

Ethics approval was obtained through the Departments of Defence and Veterans’ Affairs Human Research Ethics Committee (No. 420/22) and Flinders University Human Research Ethics Committee (No.5577).

### 2.1. Theoretical Research Framework

Phenomenology was chosen as the theoretical frame to inform this study because of its focus on exploring how human beings make sense of and transform experiences both individually and as shared meaning [40]. In particular, Heidegger’s ontological perspective was that lived experience is an interpretive process [41]. Heidegger’s conception of humans as being-in-the-world stressed the importance of the social and cultural context in which people find themselves [42]. Hence, an interpretive phenomenological approach was used to give value and emphasis to the subjective meanings, experiences, and perspectives of women veterans, seeking a deep understanding by interpreting the meaning that interactions, actions, and objects within their social and cultural context (as women in the military and as veterans) have for them [43]. This approach is about interpreting and understanding their lived experiences as human beings [44].

### 2.2. Study Population, Recruitment, and Ethical Considerations

We focused the recruitment on women veterans who had separated from the ADF since 2001. This timeframe reflects the potentially protracted nature of transition and the development of mental health and wellbeing issues, disclosure and help-seeking, and formal recognition by services and systems, including compensation entitlement for harms arising from military service. It also reflects the cohort of women veterans who have served since the campaigns across the Middle East Area of Operations (MEAO) alongside male veterans in deployed roles. Two participants who separated from service prior to 2001 were also included given their strong wish to be heard and contribute to the research. All participants were over 18 years of age at the time of interviews. They were recruited from across Australia, primarily through veteran-focused research organisations (Open Door and Military and Emergency Services Health (MESHA), Defence Force Welfare Association, Cambrian Executive), and through informal networks of the seven VPRG members. We recruited participants over a 5-month period. These approaches were important, given known issues with trust and the importance of relationships and time in coming forward for the women veterans. We did not use a formal clinical screening tool to assess eligibility; instead, we gave primacy to self-report and participants’ own judgement about providing consent because we wanted to respect the women participants’ autonomy in coming forward to share their experiences, recognising that this can be a long process for them.

Participants served in the Australian Army, Royal Australian Navy (RAN), or Royal Australian Air Force (RAAF) and included all ranks and job roles. Some participants had been on operational deployments (warlike and/or peacekeeping). Each participant was provided with a participant information sheet outlining the purpose and aims of the research, what their participation would involve, a consent form, and information regarding contact and support services if required. They could ask questions about the research prior to proceeding with the interview. They could also cease the interview at any time, ask for the audio-recording to be stopped, or choose not to answer particular questions. They were free to withdraw at any stage, understanding that once analysis was conducted and data integrated into themes, removal of individual data would be difficult. Identifiable and personal data were removed from the transcripts, and pseudonyms were used to maintain confidentiality and anonymity.

### 2.3. Data Collection

Interviews were conducted by the project officer, using Zoom online meeting software (Version 5.12.1) and a telephone, with mode, time, and date negotiated between each participant and the interviewer; one interview was conducted face-to-face. Interviews were audio-recorded, and notes were taken by the interviewer during and/or immediately after the interview to capture their initial reflections and any pertinent contextual detail. Interviews followed an open, semi-structured format, exploring experiences before, during, and after military service. The length of the interviews was determined by the participants, again to respect their autonomy and privacy, given the topic at hand. Several interviews were approximately an hour in length, some were slightly less or more, and one interview was 2.5 h long. At the end of each interview, the project officer checked with participants whether they had said all that they wanted to say; all confirmed that they had. Following the interview, participants were invited to view and verify their transcribed interview for accuracy, offer any further reflection, and make additional comments if they wished. No participants chose to do this. As interviews progressed, the interviewer and research team’s core data-coding group and project lead (three researchers) met regularly to reflect and consider the issues arising from the data and check the adequacy of the Interview Guide in giving space for participants to provide rich accounts of their experiences and perspectives. This iterative process supported the team to determine when saturation had been reached, as a group. Interviews were transcribed verbatim by a professional transcriber. Saturation was achieved after approximately 20 interviews (see Box 1 for Interview Guide).

Box 1Interview Guide.
Demographic dataLife before joining the Service and reasons for joiningExpectations and actual experience of being in the ADFViews on gender and military serviceExploration of ‘fitting in’Reasons for leavingExperiences and use of support for transitionExperiences for the first few months of transition and beyondExpectations and identity as a veteran and as a woman veteranImpacts of service on mental health and wellbeing


### 2.4. Data Analysis

We used latent thematic analysis methods, focusing on interpreting and theorising about underlying meanings in the data [45]. Data management was supported by NVIVO 12 software [46]. Data analysis was performed initially by three research team members (data analysis team) who open-coded a sample of interviews independently, then met to establish and agree upon a draft-coding plan, after which the project officer member of this team coded further interviews. The data analysis team met regularly to provide critical insights and discuss the emerging ideas. As new tentative themes emerged, earlier coded transcripts were reviewed, supported by interviewer fieldnotes and robust analysis team discussion of the data. The interpretive analysis followed van Manen’s methodological approach [44] and involved each member of the data analysis team reading transcripts and fieldnotes several times, going from parts of the text to the whole using detailed line-by-line and holistic approaches, reflecting on each interview as a whole. This approach enabled them to look in detail at description, use of language, emotion, concepts, silences, and gestures. They then presented the draft coding and interpretations to the wider research team and VPRG for robust discussion, before finalisation of themes [45].

## 3. Results

The larger study from which this paper is drawn found four key themes and various subthemes arising from participants’ experiences (see Box 2 below). More detail about themes 1, 2, and 4 is reported in a previous paper [34]. Theme 3—‘Gender and consequences of disempowerment: vulnerability to abuse’—is reported in more detail here in order to provide a more dedicated focus on this significant finding, with direct de-identified quotes to exemplify ideas (noting pseudonym, age, and length of service for each quote).

Box 2Themes from the larger study.
*Fitting in and managing identity within the military (Theme 1)*
Reasons for JoiningInitial experiences
*Gender-based challenges in conforming to a masculinised culture (Theme 2)*
Proving themselves ‘worthy’Assimilation and compromise as a survival mechanismDistancing from and ostracising other womenCaring for others: compromising caring valuesPower and gender discrimination
*Gender and consequences of disempowerment: vulnerability to abuse (Theme 3)*
Misogyny, sexual harassment, and assault: women seen as ‘the problem’.Being a mother has no place in the militaryExperiences of gender-specific health issuesExperiencing mental health issues
*Separation and transition: Being invisible as a woman veteran in the civilian world (Theme 4)*
Transition and the importance of preparation and supportAdjustment, disconnection, and the invisible veteran


### 3.1. Demographic Characteristics of Participants

This study included 22 women veterans, drawn from all states and territories in Australia. Participants ranged from 27 to 72 years in age (mean = 47 years; median = 50 years) and served in the military between 1974 and 2022. Of importance here, over half of the participants (N = 13) joined the military before the age of 20, and five of these enlisted as children under the age of 18 between 1985 and 1998 (enlistment age ranging from 16 to 44 years; mean = 21 years; median = 19 years). Over half of the participants (N = 13) served in the Army, with length of service ranging from 6 months to 37 years (mean = 14 years; median = 12 years). Ranks ranged from recruit to senior officer (actual rank might identify participant), with diverse roles that included human resources, Defence welfare, operational supply, aviation, senior command, intelligence, engineering, and nursing. Six participants left the military with the rank of officer, eight separated as non-commissioned officers (NCOs), and seven as ‘Other’ ranks. More than half of the participants (N = 14) were partnered and had children (ages ranging from infant to adult). The majority of participants (N = 17) disclosed current mental health conditions, with 12 having a diagnosis of Post-Traumatic Stress Disorder (PTSD). All participants except one reported physical injuries from service which included back, neck, shoulder, hip, and knee injuries (see Table 1).

### 3.2. Gender and Consequences of Disempowerment: Vulnerability to Abuse

A major concern reported by women veterans in this study was their real and perceived disempowerment within a dominant and masculine-driven culture, in which they had less power and were perceived to have less value, as described in more detail elsewhere [34]. The consequences of this were that women were sidelined and were the targets of sexual harassment and abuse (i.e., MST). Four subthemes are described in more detail below to demonstrate the forms that this disempowerment and its consequences took. These related to experiences of misogyny, sexual harassment, and assault; responses to being a mother; responses to having gender-specific health issues; and responses to having mental health issues.

Please be advised that readers may find some of these accounts distressing.

#### 3.2.1. Misogyny, Sexual Harassment, and Assault: Women Seen as ‘The Problem’

While there are examples of the subtle ways in which women are ’othered’, narratives revealed that women who do not conform or who challenge the values embodied in the culture are forced out or leave voluntarily. In particular, participants described more overt challenges, such as sexual harassment and assault, difficulties in balancing parenting with military demands, and challenges with (predominantly gender-based) health issues. The majority of participants described their experiences, which ranged from misogyny to rape. In all descriptions, they highlighted a military system that protects men as a more valuable resource and rejects women who report abuse.

Common to all accounts was the suppression and invalidation of these experiences as a cultural norm. Where the abuse was reported, participants revealed the traumatising nature of the military response, the protection of the male, the blaming of the victim, and stigmatising or labelling them as a ’troublemaker’ and a ’problem’ to be gotten rid of. Women who complain are perceived as no longer ’fitting in’ and adhering to the values of loyalty and conformity.

[Describing the protection afforded to a male officer by a civilian Defence employee] *So I remember working with a major at (name of unit) back in 2017. And being told that ’don’t get upset and shake your tits around, it’s not that bad’.…And I was like; did I just hear you say what you said? And it was in front of an [unit] staff member, an actual social worker. And I looked at this woman, and I said, ‘did I just hear him say that? Can you please be my witness?’ And she said, ’oh no I’m sure he didn’t mean it’. So, I thought, well there you go, nothing’s really changed. They’re still out there. There may not be as many. But they are still out there. And they are still getting away with it.*(Gail, mid-50s, 30+ years)

[Describing how she was subjected to a denigrating experience by a senior officer in 2010] *I was made—you know in the field phase when you are doing the learning the field signs, I was made to do like a hand job motion, and I was picked out specifically as a woman to do that which was obviously pretty humiliating doing it in front of like 35 men.*(Anna, early 30s, 10 years)


*So, the other thing too if you did complain—seen as a troublemaker. And particularly at sea if—You would get sent home. And often both of you would get sent home. But there was that attitude that it was the girl’s fault. Always the girl’s fault.*
(Pauline, mid-50s, 20 years)

Experiences of sexual harassment and assault were frequent, despite the era of service.


*I don’t know any women who have been in Defence as long as I have who don’t have a story about sexual harassment.*
(Heather, early 50s, 30+ years)


*Every single person I know, most of them have been raped.*
(Belinda, late 30s, 7 years)


*I haven’t met a female in Defence yet who hasn’t been assaulted in some way, shape or form.*
(Madison, late 20s, <2 years)

Six participants described being raped by peers or senior officers. Careers were threatened to suppress reporting, and, for some, this suppression of the rape and their own emotional responses became normalised. Eleanor described being physically assaulted and being raped more than once during her 20+ year military career but, having reported the first assault, she learnt that no action is taken, and this became her response with subsequent assaults, to have the career she had wanted since childhood. She reported the final assault to her senior officer, a doctor, but it was again suppressed:


*Yeah, so that was…a lot of compounding incidences that … you know, like six years later I was raped by a nursing officer that I worked with, and then three weeks later by another medic whilst I was on exercise, and I fell pregnant to the rapist…it was all covered up…so there was rape, and there were bashings, and there was things like that that had compounded over years…you know, when you’re working with doctors and nurses, and you know, they’re supposed to be held in such high regard in society, and they are, and they do a fantastic job. But for me, it was—I was one of their own, and they betrayed me…you’ve got your commanding officer that does that and covers things up. You just get so immune to everything that you’re just numb. I was just numb for years.*
(Eleanor, early 50s, 25+ years)

Katrina also had her experience invalidated by her supervisor when she reported being raped by a member of an allied military in the mid-2000s. She was blamed for being raped and informed that reporting the assault would impact the entire team and their careers, thereby “breaching the code” of loyalty to the team. Katrina commented that reporting rape to the military police meant being told that “you’re fucking someone’s entire life, their career”, again reinforcing the perception that men are valued more than women. Katrina experienced further sexual harassment during her career and commented on the extent to which women had their experiences of sexual abuse suppressed.


*He then just basically went on a sort of tirade about, a tirade if you report this then this is going to cause a diplomatic incident, we’ll be removed from the ship, you’re going to forever be known as the girl that got raped by…Everyone in the category’s going to know… just probably never going to get deployments like this again. You’re going to fuck it up for everyone…None of us knew what had happened to each other. We all knew that we were on depressants, anti-depressants or that we had PTSD, but we didn’t know exactly what. We knew that there’d been bullying and abuse, just we didn’t know just how bad it was.*
(Katrina, early 40s, 10+ years)

Quinn disclosed being sexually assaulted early in her career by a senior officer and having her career threatened if she made a report. She subsequently suppressed the assault during her nearly 20-year career and afterwards, until very recently. As with Katrina, her description reveals how common and isolating it was for women to suppress sexual assaults and thereby the psychological impacts, to have a career.


*So, in my case what I was told was it’s your word against mine, and if you open your mouth this will be the end of your career, and I’m guessing that other people were probably told something similar because I never knew there was anybody else… I just had no idea that there lots of us. I thought it was just me. Like it wasn’t spoken about…I was petrified; I was asked if I was ever angry and all the answers were no. No, I wasn’t angry, I was petrified. Q: How old were you at the time? A: 18.*
(Quinn. early 50s, 20 years)

Naomi also described being raped at age 18 but reported it to a psychologist as part of seeking mental health support rather than to the military police. She described how her medical confidentiality was breached (a pervasive sub-theme throughout the narratives), and she was subjected to public humiliation for ’betraying the code’, labelled as a whistle blower who is not part of the team and is effectively an outsider. Naomi’s description highlights the additional trauma she was exposed to through further abuse. During her 4-year career, Naomi was subjected to reputational abuse as a ’slut’, sexually objectified, bullied, ostracised, and labelled as ’crazy’ before being administratively discharged.


*I was chained up like a dog…and he put a sign around my neck, ‘don’t talk to me, I talk too much’. Back then I didn’t know what was happening, but now as a woman, oh that’s what—because I reported, I was talking to people.*
(Naomi, early 50s, 5 years)

Likewise, Sarah described being raped at age 16. She reported the rape and was then blamed, bullied, and ostracised by her peers and senior officers. She described feeling intimidated on base. Her work was sabotaged; she was set up to fail and administratively discharged within 6 months. Her description also highlights the distancing by other women who blamed her for the restrictions she put on them.


*So in the end I reported…so I became a problem and I ended up being like really ostracized…Basically by me reporting it split the women, so I am living with women that hate me because everything was tightened up because I got myself raped… didn’t know from one minute to the next if I turned a corner or walked out of my dorm or if I was anywhere on base lining up in the mess was fucked. That was the worst because that was just relentless bullying for the whole time…I would walk around, and I would get spat on by groups of men…life on-base was horrific because I couldn’t go anywhere safe or I didn’t have support with half of the women, and the other couple that didn’t fully hate me didn’t really want to be around me. So, it was incredibly isolating.*
(Sarah, early 50s, < 1 year)

While both Sarah and Naomi were viewed as “problems” and gotten rid of, others were posted out, effectively removing them rather than the perpetrator. In this way, the female was valued less and removed, while the male resource was protected. Olga disclosed experiencing sexual harassment by a senior officer early in her career, highlighting the protection of males and the removal of the victim when her assault was reported.


*He was meant to be my immediate supervisor… it was just shocking what I went through, and the people that were there that witnessed it did not stand up to protect me whatsoever… I actually broke down, it was 12 months almost and I just couldn’t handle it anymore and I broke down and confided in a friend, and I asked her not to take it any further, but sadly she couldn’t hold onto it herself, and so she did get investigations carried out with the police, the RAAF police…the decision again from defence force was to post me out away…So I feel that I was treated like the perpetrator and they protected the boys club.*
(Olga, early 50s, almost 20 years)

Linda, as a senior officer, described how she would advocate for women who had been sexually assaulted, but her description reveals a system where the male with rank and experience would be more likely to be believed and protected.


*I had to manage a lot of situations where junior (service member) would have been sexually molested…possibly sexually assaulted, given a hard time by my senior (service member)…harassed, molested, assaulted by a senior (service member) who had been in the (service) for many years who had loads of credibility, who had several commendations and it was very hard for me and the officers to manage them because you’d be going to a more senior male saying look this guy has done this ‘oh he wouldn’t do that’…so I suffered the consequences of that abuse, my deployment got delayed… Again, he was there doing his job, no consequences.*
(Linda, late 60s, 25+ years)

While Linda’s experiences occurred prior to 2000, Victoria medically separated from the military in 2022 after experiencing several years of ongoing sexual harassment from her senior officer. She described how she was held back from a deployment while her case was being investigated and again how the male was valued more as a resource despite his behaviour. Her very recent experience illustrates that the military culture remains resistant to change.


*The findings came down that I was too stupid and that I had no idea what this powerful, very qualified and experienced man expected of me as his troop, and that I was obviously needing the attention, and that there were unsubstantiated claims despite the whole squadron backing me up. And there were other things he did as well, so reports came at him from all angles, he got off Scot free, and they said, ’you know we did offer initially to protect you if we found that he was liable for something, but seems not the case, you can still post [with] him, he can still be your boss’ and case solved…And they told me that essentially the stick for when you don’t rape someone is too short, that I would have had a better case if I let it progress to rape…That was 2018.*
(Victoria, mid-30s, 5+ years)

#### 3.2.2. Being a Mother Has No Place in the Military

Being valued less than a male in the military was a perception also illuminated in participants’ descriptions of managing pregnancy and motherhood while serving. Again, experiences were diverse, but for several, becoming a mother was felt to contravene the normative male standards of what a soldier, sailor, or aviator should be, and this ’breach’ was expressed in various ways and degrees through hierarchical power and control. Descriptions revealed negative gendered attitudes towards pregnancy and that women who ‘revealed their femininity’ through motherhood were valued less. For some, their loyalty to the military and their capability in serving was challenged if they prioritised their commitment to their children. Some participants chose not to have children while serving, to have a career and avoid the challenges to parenting associated with deployments and postings. Several participants recounted experiences or observations of negative gendered attitudes and stigma towards pregnancy and motherhood.


*…but certainly, if we had children, we would never, I would never have been able to have the career that I had.*
(Linda, late 60s, 25+ years)


*I got pregnant but didn’t want to have the baby at that time, because I was going … promotion course. And it was like, this is once in a lifetime. If I don’t do it now, I’ll never get to do it. So, I had an abortion…to choose my career over having a baby…if you were pregnant you couldn’t go to sea…that’s where all the opportunity are to get promoted.*
(Pauline, mid-50s, 20 years)

Anna, describing an assault on a friend who was pregnant, also highlighted a lack of medical confidentiality.


*Like one of my friends she went to the RAP so that’s like your med centre/hospital and she said ‘look I’m pregnant’, they took her bloods, she went back to work, and by the time she walked back to work her chain of command already knew and her—but it gets worse—her sergeant took her out the back and kicked her in the belly 30 times and so she lost the baby…the military will hand out abortions like nobody’s business.*
(Anna, early 30s, 10 years)

Terri recounted how she was not allowed to continue working following a service transfer at a higher level while pregnant and was offered the choice of an abortion, to continue in her career or take leave and return at a lower rank. Terri commented that she knew that this was discrimination, but that to pursue this would have her labelled as a ’trouble-maker’, which would have impacted the rest of her career.


*I was pregnant, and they suddenly said ‘oh well we can organize an abortion for you. You don’t have to worry about it’.…the base treated me like absolute filth…I was just ‘oh you know what, this isn’t worth it’.…I think that if I had pushed it to the point where I had stayed in then that pushing would have followed me…You would have been labelled as a troublemaker…you should be at home with the kids any way because that comes up too, what are you doing here? It’s very very wrong… I don’t think much has changed.*
(Terri, late 50s, <5 years)

Balancing motherhood and a military career was highly dependent on the attitude, power, and control of the senior officer. Belinda highlighted both the negative attitude towards mothers and the power and control (and abuse) that rank in the military can exert over individuals.


*One of my sergeant’s tried to kill herself at work and she had post-natal depression and they’d sent her husband away and were trying to force her to come back to work with a baby and two little kids…and my CO’s decision was to discharge her.*
(Belinda, late 30s, 7 years)

Eleanor commented on her experiences of motherhood and the bullying she was subjected to by a female nursing officer. She commented on her perception that older women, in particular, may discriminate against mothers because of their own experiences of having to fit in.


*They had it pretty tough. So, I think there’s jealousy, I think there’s jealousy in some of these older high-ranking officers, Army in particular that I’ve worked with in joint health command and in headquarter spaces. They are still of the old opinion that you know, well, women can’t do it all, you’ve got to concentrate on the military.*
(Eleanor, early 50s, 25+ years)

Anna described enjoying her career until she became a mother and, from then on, was bullied and harassed by her senior officers until a medical separation in 2019 when she was in her late 20s. She described how her identity as a mother was constantly challenged and controlled, impacting her choice of birthing hospital, breastfeeding rights, and maternity leave. Anna did not conform and, having challenged restrictive maternity policies and work demands placed on her, was labelled a ’troublemaker’ and eventually medically discharged.


*So, I was called in 58 times on my unpaid days. Again, I think it was this bullying, yeah because they knew that I had children to look after… I felt I was in a domestic violence relationship with my employer… just the constant harassment, holy Jesus…come in day after day bringing my baby, it was just not right but what are you supposed to do. You say no and you get charged, and like I have it in writing a couple of times as well that if I didn’t rock up to xyz I would get charged… Again because of my motherhood status and sort of being perceived as less than, and you know when your kid is sick and you’ve to take a carer’s day, just rolling the eyes and the fact that they make you come in with your sick kid who is on antibiotics into the workplace, like it’s just completely revolting.*
(Anna, early 30s, 10 years)

Gail described how she was supported as a single parent, highlighting the importance of having a supportive senior and a support network as well because the military will not help.


*I’ve been lucky. I have always managed to be able to have that time… Create that village, because the village is what will help you. And that’s how I survived for so long. I had a really good support network of friends who—who would help me…. If your area manager I think had the decision-making drive to do things, then it would happen. But if you had someone who couldn’t see past the desk, then it was pointless.*
(Gail, mid-50s, 30+ years)

Olga tried to meet the demands of the job and described the lack of balance between work and motherhood. Her description reveals the difficulties for women who feel they have to work harder in balancing these roles to be accepted in the military:


*It was difficult because I had to work twice as hard as the guys to get that recognition, and that meant even when children [came] along getting there earlier and staying back late at night. So, it was totally unbalanced, absolutely… my children were young I didn’t have a life, I literally was a machine. I would literally get them up and 5:30 or 5 am in the morning, get them dressed, send them to childcare, I wasn’t the mum that I wanted to be. Then I would go to work, work really hard, long hours and to come home to have to do it again.*
(Olga, early 50s, almost 20 years)

#### 3.2.3. Experiences of Gender-Specific Health Issues

Along with pregnancy and childbirth, some participants highlighted challenges in seeking health care for reproductive and gender-specific health issues. Participants’ descriptions revealed both their perception of a health system designed around responding to male health care needs and a level of organisational inflexibility in responding to gender-specific health needs of women; not being empowered to have them managed properly. At the same time, the control that the military had over the female body was highlighted by some participants.

Both Anna and Heather described lack of choice in medical care for reproductive issues. For Anna, control was expressed in lack of personal choice regarding birthing and maternity care. Anna had a home birth and described the consequences for her. Inflexibility in meeting female-specific health needs was highlighted by Anna, describing how military fitness policies failed to take account of her post-childbirth health issues, resulting in physical damage.


*I didn’t make it to the hospital with my second, and so then I was with the chaplain, and I was brought into a room with six men and sort of given a real stern speaking to, and said if I was to do that again I would face disciplinary action…the level of control that they had over my body, my reproductive body, and someone said to me the other day ‘well as soon as you found out that they were so restrictive, you know you could have just left’, but it’s like oh but you can’t, you’ve actually got to give 3–6 months’ notice…I was probably like 49 kilos and carrying like a 45-kilo pack on my back, like what’s that going to do to a pelvic floor especially when you are postnatal…those things weren’t considered and the fact that you are expected to do all of that within your first three months of return to work is in my opinion negligent really.*
(Anna, early 30s, 10 years)

Heather described her need for reproductive care post-childbirth and the difficulties she experienced in seeking this care in a system she perceived as lacking expertise but having control over her body. Likewise, Heather commented on the lack of understanding (or misalignment) of women’s reproductive health in military policy.


*Because the majority of veterans are men, the standard set of services are designed for the standard veteran which is a man…women’s health is not managed well in Defence…I had what the obstetrician called the mother of all ruptures, like I nearly died, and [baby] nearly died and it was pretty horrific and traumatic so I get my pap test at a GP, it’s actually has to be done by a specialist, and Defence—I had a really hard time convincing them…they are like ‘just go to the women’s health nurse at the local Army health centre’, and I am like ‘no I am not going there because’…I had to justify it and there was these hoops to jump through…I think coming back from maternity leave and particularly with me who had had such a—both of my children were caesareans but [name of baby] was so much more—there was just so much more damage. Again, you had to justify having longer before doing your fitness test again.*
(Heather, early 50s, 30+ years)

Gail described how her gender-specific health issues were the subject of discussion at the welfare board meetings where decisions were made about medical fitness to continue. She highlighted the difficulties that the male-dominated board had with discussing women’s health.


*My OC—a major knew of my pelvic floor issues. And every month I would have to sit at a table involved in what they call a unit welfare board. And because of my rank I always asked to attend… I would sit at a table with about 12 to 20 people—doctor, psychologist, the commanding officer, the RSM, your OC, your 2IC, your unit doctor, your unit rehabilitation consultant. You might have a member from DCO there. You might have the Chaplain there. So, there was a whole gamut of people around the table. And they would talk about what your medical issue was. And as a 54-year-old woman, to sit there, and have my pelvic floor discussed…everyone would be very embarrassed by it…I didn’t give a shit in the end. And I would just say how it was. ‘Yes, I’m struggling. Yes, I’ve got the surgery coming up. Yes, there are days when it’s worse than others. Yes, I’m 54. Yes, I expect that I’ll be going through menopause soon’…Instead of me being made to feel uncomfortable, I’d put it back on them. I thought, no bugger you. If you want to know what’s going on with me, I’m going to tell you.*
(Gail, mid-50s, 30+ years)

Other participants revealed how they were no longer perceived to conform to military standards because of their gender-specific health issues and were either medically discharged or encouraged to leave voluntarily. Ingrid described how she loved being in the military until she underwent a hysterectomy for cancer, then was bullied and harassed by her senior.


*It was a very positive life in the Army for me, until I had stage 5 cancer and then I get treated like a leper… I think I was discriminated. You know, I was told to drive to work… and I’d just had a hysterectomy and they expected me to drive from—it was roughly 30 Kms just to sit down at my desk and do nothing. So, I had top secret clearances, and I really wasn’t allowed to do anything.*
(Ingrid, early 50s, 10+ years)

Similarly, Katrina described being bullied following a mastectomy, and Frances was also medically discharged, having been deemed no longer fit to deploy following treatment for breast cancer.


*…a full bilateral mastectomy and breast reconstruction, after that, I was bullied into a mental breakdown…officers who were bullying me because of my rehab plan, even though they’d said, one of them said his wife had gone through a mastectomy and they’re we’ll be so supportive, flexible as rubber bands and all this sort of shit and then they were absolute cocks. And I was just like ‘this is great’, I went in to get my breast cut off…I lost a large amount of upper body strength as a result of that. But then I was bullied into a mental breakdown over a seven or eight-month period because of my rehab.*
(Katrina, early 40s, 10+ years)


*They discharged me because they said having had breast cancer, I was non-deployable…that was an awful shock…it had been my life. It had been everything I’d ever wanted, and I was loving what I was doing, and I was good at what I was doing…and then to be told that I was no longer able to continue because they said, ‘you’re not deployable’.*
(Frances, early 70s, 20+ years)

#### 3.2.4. Experiencing Mental Health Issues

Some participants commented on their perceptions that the health system is used to exclude women with mental health issues from service. While this might also be a generalised military response to perceived ’weakness’ in an individual regardless of gender, women are considered to more commonly engage in mental-health-help-seeking behaviour earlier than men. Stigmatisation of mental health was commonly mentioned in the interviews, whether overtly or through the reluctance of some participants to initially disclose their issues or when they sought help early.


*Men sort of probably hide the fact that they need mental health, like by the time they need help like they are literally on the verge of suicide half the time, like they’re struggling with PTSD and that’s when you first hear about it. With females, like they’re more open with the fact that they’re struggling before it gets too far, and they try to seek help, but the problem is when we as females start seeking help early, we really do get shoved aside like quite quickly.*
(Deirdre, late 30s, 5 years)

Madison described her perception of the public and organisational stigmatisation that takes place when psychological support is sought in the military. For Madison, this was a barrier to seeking help for her mental health issues.


*My experience and the experience of many others again … you go into medical and you say you are having trouble then you’re immediately downgraded and generally medicated in some way, shape or form, and if you are put on any kind of medication that’s an automatic fine for unfit to handle weapons, and you can’t be the only person in the unit that doesn’t have a weapon because everybody knows why you don’t have one.*
(Madison, late 20s, <2 years)

Eleanor described how her help-seeking over the years to deal with the multiple instances of sexual abuse, bullying, and harassment was used to medically discharge her, highlighting again military institutional abuse in the use of the health system by command.


*It was the doctor that was bullying me decided that I should go for a psych evaluation, and I said, ‘yeah, sure, I’ll go for a psych evaluation, let’s sort this out’. Unfortunately, the psychologist that they sent up decided after speaking to me, after an hour, that I should be administratively discharged…This was a Defence Major psychologist…he’d gone through all my records, and he’d written my whole life out to my commanding officer, and he was not entitled to see that information, it had nothing to do with my work. It was all about assaults and how much time I’d spent in with psychologists, that I’d had extensive psychological care, and I should be—you know, I should be cured by now because I’d spent six years with a psychologist, and that it wasn’t worth spending the resources on me, pretty much because I’d already exhausted beyond my fair share.*
(Eleanor, early 50s, 25+ years)

## 4. Discussion

The experiences of discrimination, disempowerment, and abuse described by the women veterans in this study had significant impacts on their perceptions of self-worth, confidence, safety, shame, and so forth. This sense of betrayal by the military then had profound impacts on their transition and mental health and wellbeing beyond military service.

The sexual abuse of women in the military is a persistent finding in contemporary international research [12,15,16,17,27,28], and this study indicates that it is also a continuing concern for women who have served in the Australian military. Despite diversity in service experiences, career length, or rank, participants in this study experienced common challenges in their military service and transition to civilian life that were highly gendered. Aligned with the theoretical frame of phenomenology and its focus on the interpretive process, this study provides a detailed account of the subjective meanings, experiences, and perspectives of an Australian sample of women veterans, seeking a deep understanding by interpreting the meaning they gave to the social and cultural contexts and challenges as women in the military, and as veterans, that these experiences have for them [43]. Prominent among these challenges was the experience of MST, underpinned by a masculinised military culture that enabled, ignored, and condoned it.

### 4.1. Masculinised Culture

Challenges arising from a masculinised military culture are commonly reported in research across the Five Eyes countries in relation to the experiences of women veterans [12,15,16,17,18,19] and have been discussed in more detail elsewhere [34]. Of relevance here is how this culture creates the conditions for gendered discrimination and MST to occur and also the conditions for how it is responded to during military service and transition and how its impacts are felt and responded to post-military service.

In the current study, several women veteran participants described how they distanced themselves from feminine behaviours, such as their efforts to excel with physical fitness and vigilance against any indication of weakness. These behaviours are commonly reported in the literature [19,47] as the means to integrate more successfully into the masculinised culture [48], and they were evident in how participants described the advice they were given by (male) relatives prior to joining and in reflections on their own integration strategies. Conversely, when women veterans tried too hard to fit in, they were seen as incompetent and weak or labelled as bossy and dominant and a threat to male dominance, leading to stigmatisation, ostracisation, and bullying of women who do not conform to normative male standards by other women (findings reported in other research [17]).

Whether in traditional caring roles as nurses or in leadership roles, participants in our study described how they were challenged by the hyper-masculine military culture. Previous research, for example, a US study [12] found that women in healthcare positions were treated differently and with more equality than those in operational units; however, we found that women in healthcare roles in the military still experienced gender-based discrimination and abuse. We also found that gender-based discrimination was not more pronounced in male-dominated trades or units [28]; it was pervasive regardless of what role the participants played in the military. We found that positional power played a component role in the positive experiences of some women participants, for example, being in charge of personal leave of other personnel. It is unclear, however, whether and to what degree this was based on perceived or actual male positional power being involved, for example, the woman service member being ‘off limits’ due to being married to a male senior leader.

### 4.2. Sexual Abuse

Almost all 22 participants in this study described challenges during their military service, which they perceived to be because they are women. Their most pervasive experiences were of abuse. While institutional abuse, such as bullying, and medical and administrative abuse, such as exclusion and invisibility to their needs as women, were common, they perceived that sexual abuse was most pervasive and impactful on their service and transition to post-military life. Researchers consider military sexual abuse to be a large problem globally in militaries, given that ’martial masculinity is inherently sexualised in aggressive ways’ [28] (p. 89). There is now extensive research literature on the health-related and psychosocial impacts of MST, including PTSD, suicidal ideation, suicide attempts, alcohol and drug dependence, relationship difficulties, homelessness, and reduced quality of life [16,49,50]. In concordance with the international literature, the current study revealed difficulties for several participants in all domains of their post-military life.

While there is very limited previous Australian research with women veterans on their experiences of sexual harassment and sexual abuse whilst serving, this issue has been brought to light by numerous Australian governmental investigations and reviews in recent decades, which indicate that it is an endemic issue in the ADF [22,24,51]. It has also been acknowledged as a significant concern in other countries with similar military institutional arrangements, such as the UK [38,52,53], Canada [54], and in the US [55]. Of note though, the very recent US Veteran Affairs report [55] used the term ‘forced unwanted sex’ throughout its reporting, suggesting that this military institution continues to minimize sexual abuse of women veterans. Likewise, in the UK, evidence of deeply ingrained cultural attitudes that continue to minimize military sexual assault have been highlighted by the government’s refusal to have MST officially recognised [52] and in the disproportionately high rates of women veterans who do seek help for MST being diagnosed with emotionally unstable personality disorder (EUPD) [53,56]. Roberts [56], drawing on evidence from Edwards and Wrights’ UK report [53], stated, “In 2022, out of 393 women referred to Salute Her, 133 women had been diagnosed with EUPD with damaging consequences for their civilian lives. Its symptoms include intense negative emotions such as panic, shame, terror, impulsive behaviour—self harming, for instance and having upsetting thoughts. To me, that behaviour is a perfectly natural response to what many of the women have experienced, yet they end up pathologized” (web resource, no page number).

The finding that the military institutional culture invalidates the experience of sexual abuse, protects the male, and suppresses reporting is consistent with international research [12,17,19,47]. Daphna-Tekoah et al. [27] found that this invalidation was dominant in a study with US and Israeli combat women veterans along with suppression of the crimes and silencing at personal, unit, and system levels. Consistent with our study, researchers have reported that there are military-specific obstacles to reporting sexual abuse and seeking justice, such as shame, fear of loss of career, and being seen as responsible for breaking unit cohesion and destroying the careers of men [12,28,47]. Bourke [28] (p. 93) commented that in, “a workplace that valorizes group cohesion, loyalty and stoicism, ostracism is a powerful inhibitor of speaking out against brothers in arms”. Burkhart and Hogan [12] reported that military culture, premised on gender power relations, can lead to blaming the victim, labelling them as a troublemaker if they report the abuse, a finding made in the current study.

The long-term social, emotional, and psychological impacts of this “second betrayal” by ”the military family” for some participants in this study were profound; the impact was reinforced by the complete contrast with their positive reasons for joining and commitment to service and careers, which is also consistent with other research [50]. Recent Australian research into military institutional abuse has described how survivors (men and women) struggle to move on because of this betrayal and characterize the trauma as an identity wound [57]. Consequently, perceptions of institutional betrayal have been found to lead to premature separation from service and long-term negative health consequences, including more severe symptoms of depression and PTSD along with increased suicide risk during or following military service [16,47,49,50]. Obstacles to reporting, invalidation of the abuse by others and protection of perpetrators effectively silences and isolates the victim and prevents them from connecting with each other [48]. For some of the women in our study, this silencing resulted in years of managing the psychological impacts of the abuse alone, internalising feelings of blame and shame, unaware that they were not alone. Highlighted as ’different’, some participants had described being socially stigmatised and rejected by other women in the military, also increasing feelings of isolation.

Despite the extent of the problem of abuse within the military, the extensive and predominantly US-based research focus has been primarily on treating the ensuing abuse at the individual level rather than analysing the military institutional culture that enables abuse to occur [28]. In this way, the individual’s response to abuse is pathologised and addressed (or not) only at the level of the individual. As Bourke [28] (p. 102) describes, “On the one hand, victims of sexual assault in the military are regarded as the embodiment of weaker femininity…this assumption is strongly resented and resisted by servicewomen… On the other hand, their identity as female members of a ‘warrior caste’ forces upon them the notion of agency: they are young, active, and strong. The issue of female agency is routinely used in civilian court cases to argue that women are unrapeable: the fit, combat-trained military woman is even more so. They become responsible for their own victimization”.

The key resource for therapeutic treatment of trauma in the veteran population in Australia only makes brief reference to sexual abuse as a traumatic exposure that can occur through military service [58]. There is limited recognition that sexual abuse in the military can differ markedly from civilian experiences. There is no reference to the masculinised military culture that allows sexual abuse to occur or the fact that this abuse occurs in the workplace where the military has 24 h control over the individual, the power inherent in rank and the command structure, or the ’code of silence’ described as a key pillar of institutional culture [57] that silences or punishes reporting.

### 4.3. Invisibility in Veteran-Specific and Broader Health Services

Several participants described negative, adversarial experiences of seeking support, recognition for physical and/or mental injuries, or compensation through the DVA, consistent with previous Australian and US research [14,27]. In particular, they described significant challenges with help-seeking for their mental health and wellbeing once they left the military. For some, this mirrored the military culture in which their MST occurred and triggered further trauma, re-victimisation, disempowerment, and shame, particularly if they were not believed then or now [36]. A US study [59] reported that women veterans were inclined to hide their military service experiences rather than seek to justify eligibility for support services once they transitioned. Crompvoets [14] emphasised how reluctance by some women veterans to embrace their veteran status had implications for them accessing existing veteran support services. Participants in our study described silencing or downplaying of their veteran identity due to feelings of shame and social disconnection resulting from military institutional abuse, with potential for their veteran status remaining unrecognised, unacknowledged, and their needs unaddressed over many years. Their help-seeking experiences also suggest that Australian post-military health and mental health services may not be well-equipped to recognize, engage, understand, and support women veterans.

Participants in the current study disclosed a high prevalence of mental health diagnoses due to their military service, particularly PTSD, consistent with other research [17,47,48,60,61]. However, whilst women veterans can experience similar barriers to help-seeking as men veterans, particularly concerning self-stigma, it may be that women veterans’ experiences differ due to the additional work performed by women veterans in mitigating any signs of physical or emotional weakness in the military [48] and due to high rates of gendered harassment, discrimination, and abuse whilst serving. A US study [62] exploring gender differences in trauma types and themes in veteran PTSD found that many of the women veterans involved had developed PTSD after sexual assault. Women’s reactions emphasised mistrust, betrayal, and anger, whereas combat-related reactions such as survivor guilt, moral injury, and horror were implicated more often to explain men veterans’ PTSD [62]. These examples of emerging research suggest that more needs to be understood by health services about gender-specific trauma experiences and mental health support needs of women veterans.

In addition to formal health service support, we know that women veterans value social connection and peer support from other veterans, post-military service [12,14], to minimise negative mental health and wellbeing outcomes during transition [16]. For women who have experienced discrimination, abuse, and MST during military service, these supports provide opportunities for the telling of their stories safely and for validating their experiences [14,15]. Previous research has found that some women veterans, especially those who have experienced MST, feel unsafe when they have reached out to veteran-specific services [16,48]. Australia’s Veterans and Veterans Families Counselling Service (known as Open Arms) has been slow to respond to the needs of women veterans. Despite an expressed desire by women veterans for groups tailored to women [61], Open Arms have reported logistic issues and low numbers as barriers to establishing women-only services. A recent review of evidence for group-based programmes for women veterans [36] found that a high number of studies involved very small samples and were of low-quality, indicating a need for more research in this field. That review found that strengths-based women-only groups, facilitated by women, that created safe spaces for women veterans to share their experiences, enhanced self-expression, agency, and self-empowerment, and these conditions were particularly important for women who had experienced MST [36].

### 4.4. Implications for Policy

The findings of this study suggest that a range of policy shifts are needed to inform improvements in the service and transition experiences of women veterans. The final report of the Australian Royal Commission in Defence and Veteran Suicide (RCDVS) made many recommendations [22]. In response to its findings, the Australian government has committed to implementing the majority of the recommendations. Notable actions include addressing sexual misconduct by implementing measures to discharge ADF personnel convicted of sexual offences and tie officer promotions to ‘emotional intelligence’ in order to foster a safer and more supportive environment for all service members, particularly women. Australia’s DVA has also committed to undertake specific initiatives to support women veterans. These include the development of a National Women Veterans Strategy [63] to focus on funding and designing services and systems to better recognise and respond to the needs of women veterans (including the ADF and DVA institutions themselves, as well as mental health and broader health services in the community, and ex-service veteran support organisations). They also include the strengthening of the National Women Veterans Policy Forum that was established in 2016 to provide a platform for women veterans to directly communicate issues to the government and DVA [64]. The RCDVS report [22] also pointed to the need to use data more effectively to research issues across the ADF (military) and DVA (veteran post-service) divide to drive improved outcomes for women veterans.

### 4.5. Limitations, Strengths, and Suggestions for Future Research

This study has a number of limitations. Whilst it involved a small heterogenous sample of 22 women veterans with diversity in service type, era served, length of service, operational experience, age, rank, and role, the focus of this qualitative phenomenological study design was on a thick description of lived experience not representativeness. Also, this research did not focus on potential changes in the cultural experiences of the military over time, which could be explored further as part of an anthropology-focused study. A further limitation was that the sample was relatively homogenous, with the majority being white, and sexual orientation or LGBTQIA+ status was not identified.

Given the breadth and depth of what participants shared, we believe the small sample size might be due to their readiness to disclose and talk about MST and other potentially traumatic experiences that occurred during their military service. They may also experience social isolation and not identify as veterans, given the dominant masculine military environment. A further potential limitation of this study is that women veterans with positive experiences of military services may be under-represented. Our study and the recruitment materials explicitly sought women veterans with any experiences (positive, neutral, or negative); women with positive experiences may have been less interested in talking about their experiences.

A strength is that interviews conducted via video call enabled us to hear from women situated across Australia, afforded them privacy and anonymity when discussing significant taboo and sensitive issues ‘virtually’, and enabled participation access for women who may have had difficulty leaving their home due to poor mental or physical health or other circumstances, such as employment or parental duties. A limitation of ‘virtual’ interviews is that it may have limited the ability to observe body language, develop trust in the interview process, or gauge participants’ distress. However, the richness of the collected data suggests this was not a limitation in this study.

An overall strength of this study was the in-depth nature of the interviews. Future research could focus on gendered experiences for women veterans in different service corps in the ADF, by different occupations, their operational deployment experiences, and more contemporary experiences of service and transition. Further research could involve comparative interviews with military leaders at all levels (e.g., non-commissioned officers, junior officers, senior officers) about how they have tried to combat these issues of gender discrimination. Further research could also examine the occurrence of military sexual assault for both women and men veterans, given the masculinity culture in which it occurs is pervasive and remains largely unaddressed for both groups [57]. For example, Bourke’s [28] review found that, in 2010, 46% of all veterans who had been diagnosed with MST were men.

## 5. Conclusions

Women veterans face significant challenges whilst serving in the military and in transition, due to the highly masculinised military environment, which is largely replicated in transition. Some women veterans experience significant gender-based barriers to services and support. The experiences of harassment, discrimination, and military sexual assault (and the consequent MST) for the Australian women veterans who participated in this study were described as largely stemming from this entrenched gendered military culture. These experiences for women veterans are also pervasive in other countries, such as the UK, the US, and Canada.

The ADF and DVA have recognised the need for a cultural change to address gender issues during military service and in how supports for transition and post-military service are provided. There is work currently underway by the government to develop a National Women Veterans Strategy [63]. However, military organisations have a contested and largely conservative history and, like in other countries, progress from many Australian inquiries over several decades has been slow in translating policies arising from such inquiries into practice within its military institutions and within institutions supporting its veterans in the community after service. This study adds depth and understanding to the small but growing body of research on the experiences of women veterans.

## Figures and Tables

**Table 1 ijerph-22-00584-t001:** Demographic characteristics of participants.

Characteristic	N	Characteristic	N	Characteristic	N	Characteristic	N
Service Branch Navy Army Air Force	3 13 6	Length of service (in years) Under 1 1–4 5–9 10–14 15–20 20+	1 4 3 5 3 6	Age at enlistment (in years) 16–17 18–19 20–24 25–29 30–34 35+	5 8 6 0 2 1	Rank on discharge Officer Non-Commissioned Officer Other rank Not disclosed	6 8 7 1
Age (in years) 20–29 30–39 40–49 50–59 60–69 70–79	1 5 3 11 1 1	Relationship status Single Partnered Divorced	4 14 4	Children 0 1 2 3 4 5	9 2 5 4 1 1	Health Status Mental illness from service Physical injuries from service Comorbid mental and physical injuries from service Mental illness not from service No mental or physical injuries	2 4 14 1 1

## Data Availability

The datasets presented in this article are not readily available due to privacy or ethical restrictions.

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
