# Peer review of "Australian Women Veterans’ Experiences of Gendered Disempowerment and Abuse Within Military Service and Transition"

_ijerph, 2025, doi:10.3390/ijerph22040584_

Round 1
Reviewer 1 Report
Comments and Suggestions for Authors
Review comments
Overall, this article is very similar to the one previously published about this small qualitative study of women veterans. Entire paragraphs are the same, which is unacceptable.
I am concerned about the heterogeneity of the very small study population, especially with regard to the era of service. One of the arguments is that military culture is hyper-masculine, but the era of service for the participants extends back to 1974. There is insufficient attention given to how military culture may have changed in this time period (as culture has changed in general). Given that recruitment was through established veteran organizations, how do you think the participants may be different from women veterans generally? The article makes claims about women veterans’ experiences but more effort should be given to the limitations or nuances given the small, heterogenous sample. You noted that “Saturation was achieved after approximately 20 interviews” but was that related to some of the themes from the broader study or specifically related to women’s experiences of abuse? Saturation usually requires that analysis is ongoing (i.e. as in a grounded theory approach), however it sounds like analysis was conducted after the data was collected.
You do not mention the role of the Veteran Project Reference Group (VPRG) members. How was this group involved?
What strategy was used for thematic analysis? (E.g. latent vs semantic coding, a priori identification of themes, etc.) What strategies were employed to check for the validity of the interpretations of the data. You note that participants did not want to verify their transcripts, but were there other ways that you attempted to examine the trustworthiness of the interpretations and conclusions?
In the limitations you note “we believe the reluctance of more women veterans to participate in this research” however you did not mention any reluctance when describing the recruitment and sampling. Did recruitment end because of saturation or because it was difficult to recruit participants?
You describe a strength of the study being “in-depth nature of the interviews” but I would question whether a single one-off one-hour interview is in-depth in terms of developing trust and holding space for complex stories and ideas to emerge.
In addition to these major concerns, the manuscript needs edits for grammar, typos, etc. e.g. the abstract has sloppy incorrect punctuation. There are also sentences that are incomplete (e.g. “Within such limited support structures, lack of disclosure and reporting of MST, and lack of engagement with health services to deal with the impacts of MST on physical and mental health.”)
The literature review would benefit from adding quantitative estimates for some of the experiences that are described as “common.”
The article overall could use some rewriting to better substantiate claims or add more nuance. For example, a sentence like “Support services are inherently gender biased” is not backed up by evidence and reflects a pre-existing belief that undermines the trustworthiness of the qualitative inquiry in the article.
The aim of the study, as described in the last paragraph of the introduction, is not sufficiently connected to the literature review. This is the first time we hear about suicide or mental health. There should be additional literature substantiating the focus on this. However, this paragraph is nearly identical to the last paragraph of the introduction in the previously published article, so perhaps it is not actually connected to the specific hypotheses of this manuscript. Also, as the study only included women, the aim should not be framed as a comparative one to men’s experiences (e.g. “why rates of veteran suicide are higher for women veterans”).
Reviewer 2 Report
Comments and Suggestions for Authors
1. It would be better to separate the literature review from the Introduction. At present the extant literature is too briefly presented. The reader gets an idea of the names of references but not their content. With such brevity, it is difficult to gauge the quality of the literature review.
2. In the discussion, the themes masculinized culture and peer support seem to be drawn from the larger study. This article only presents findings on sexual abuse and health issues. New ideas based on data that is not presented here cannot be part of the discussion.
3. The conclusion could present a couple of policy/institutional recommendations instead of just leaving it at having contributed to the in-depth understanding of the problem.
Reviewer 3 Report
Comments and Suggestions for Authors
I am grateful to have had the opportunity to review and evaluate a manuscript that deals with the specific topic and problem of gender discrimination in the Australian military system. Research of this type and target orientation, especially in systems such as the military or the police, carries a special weight and specific value because it deals with structures that are deeply gendered and opposed in terms of rights, status, obligations, valuation, prejudices and (non)discrimination.
The manuscript is clear, reviewed, understandable, well-conceived and structured, and the presentation of the research material has a methodologically logical flow. In terms of language, semantics and spelling, the manuscript is correct, the language is clear, and the style is good and suitable for the research potential necessary for understanding the analyzed problem. The terminology used is adequately selected and highly professional, but at the same time adapted to the standards of the average researcher who, as a previous education, has a basic academic level of familiarity with the conceptual-categorical apparatus in the field of gender studies. The structure of the work is well and logically laid out, and the relationship of the parts is such that it forms a coherent whole, which is consistent in terms of content and methodology.
The study, in its projected research design, largely achieved what it set out to do. The design and methods used are compatible with the set objectives.
In the chapter of the manuscript entitled Limitations and Future Research, the authors themselves provide an overview of the key limitations of the study. As a reviewer, I agree with the listed "flaws" and add the following:
1. The conceptual-categorical apparatus lacks a clear and concrete definition of the terms "female veteran" and "war/military veteran". It is necessary to clarify the specific meaning of both terms.
2. There is a lack of description of the wider context (gender analysis and historical genesis of Australia's security system) within which the research was conducted.
3. It would be useful to have a brief introduction to the structural characteristics of the military system in Australia (where and in which positions in the army women are engaged; what jobs they mainly perform and where they are most represented; what status they have; what types of engagements they had within the framework of war operations) .
4. There is a lack of a theoretical research framework (conceptual approach on which the target direction of the research is based), as well as relevant previous research conducted in Australia, the narrower or wider region, or on other continents, whose findings would be confirmed or compared by this study.
5. I suggest that the authors refer to the analysis of the army as a gender institution in the introductory part of the paper, guided by the research of Acker (1992, 2006), who dealt with the key determinants of the police as a gender institution and the levels of control and segregation represented in the police. In doing so, the authors can compare the specifics of the military and the police as gender systems, which would largely explain the problems and challenges faced and fought by women as veterans (war/military) in the military system in Australia.
1. Acker, J. (1992). From sex roles to gendered institutions. Contemporary Sociology, 21, 565-569.
2. Acker, J. (2006). Inequality regimes: Gender, class, and race in organizations. Gender and Society, 20, 441-464.
Round 2
Reviewer 2 Report
Comments and Suggestions for Authors
Revisions satisfactory.
Author Response
Thank you for your suggestions and support of this work.